# Predicting adverse pregnancy outcome in Rwanda using machine learning techniques

**Theogene Kubahoniyesu**[1,2]*, **Ignace Habimana Kabano**[1]

**1** African Centre of Excellence in Data Science, University of Rwanda, Kigali, Rwanda, **2** Research, Innovation and Data Science, Rwanda Biomedical Centre, Kigali, Rwanda

* theogenek9@gmail.com

## Abstract

### Background

Adverse pregnancy outcomes pose significant risk to maternal and neonatal health, contributing to morbidity, mortality, and long-term developmental challenges. This study aimed to predict these outcomes in Rwanda using supervised machine learning algorithms.

### Methods

This cross-sectional study utilized data from the Rwanda Demographic and Health Survey (RDHS, 2019–2020) involving 14,634 women. K-fold cross-validation (k = 10) and synthetic minority oversampling technique (SMOTE) were used to manage dataset partitioning and class imbalance. Descriptive and multivariate analyses were conducted to identify the prevalence and risk factors for adverse pregnancy outcomes. Seven machine learning algorithms were assessed for their accuracy, precision, recall, F1 score, and area under the curve (AUC).

### Results

Of the pregnancies analyzed, 93.4% resulted in live births, while 4.5% ended in miscarriage, and 2.1% in stillbirth. Advanced maternal age(>30 years),women aged 30–34 years (adjusted odds ratio [AOR] = 5.755; 95% confidence interval [CI] = 3.085–10.074; p < 0.001), 35–39 years (AOR = 8.458; 95% CI = 4.507–10.571; p < 0.001), 40–44 years (AOR = 11.86; 95% CI = 6.250–21.842; p < 0.001), and 45–49 years (AOR = 14.233; 95% CI = 7.359–25.922; p < 0.001), compared to those aged 15–19 years, and multiple unions (polyandry) (AOR = 1.320; 95% CI = 1.104–1.573, p = 0.002), and women not visited by health-care provider during pregnancy (AOR = 1.421; 95%CI = 1.300–1.611, p<0.001) were factors associated with an increased risk of adverse pregnancy outcomes. In contrast, being married (AOR = 0.894; 95% CI = 0.787–0.966) and attending at least two antenatal care (ANC) visits (AOR = 0.801; 95% CI = 0.664–0.961) were linked to reduced risk. The K-nearest neighbors (KNN) model outperformed other ML Models in predicting adverse pregnancy outcomes, achieving 86% accuracy, 89% precision, 97% recall, 93% F1 score, and an area under the curve (AUC) of 0.842. The ML models constantly highlighted that woman

**Data Availability Statement:** The data used in this study were obtained from the Demographic and Health Surveys (DHS) Program and can be accessed through their website https://www.

dhsprogram.com upon request. The specific dataset used is the Rwanda 2019-202 Demographic and Health Survey (DHS) women's dataset. Access to the data requires a formal request through DHS program's data request system to ensure compliance with legal and ethical guidelines. The data are anonymized to protect the privacy of respondents. The authors did not have any special access privileges to the data. Others can access the datasets in the same manner.

**Funding:** The author(s) received no specific funding for this work.

**Competing interests:** The authors declare that the research was conducted in the absence of any type of relationship that could potentially trigger any conflict of interest. All authors agreed to submit to the current journal, accountable for all aspects of the work, and provided final approval of the version to be published.

with advanced maternal age, those in multiple unions, and inadequate ANC were more susceptible to adverse pregnancy outcomes.

## Conclusions

Machine learning algorithms, particularly KNN, are effective in predicting adverse pregnancy outcomes, facilitating early intervention and improved maternal and neonatal care.

## Introduction

Maternal health is crucial for ensuring healthy pregnancies and live births. It remains a major health priority, especially in developing countries, where 95% of preventable maternal deaths occur [1]. According to the World Health Organization, an estimated 86% (254,000) of global maternal deaths in 2017 occurred in Sub-Saharan Africa and Southern Asia. Specifically, two-thirds (196,000) of these deaths were in Sub-Saharan Africa, while one-fifth (58,000) occurred in Southern Asia [2].

Every pregnant woman hopes for a safe delivery, avoiding the loss of her life or her baby's life. However, pregnancy outcomes such as live birth, spontaneous abortion, preterm delivery, or stillbirth are not always predictable by expectant mothers [3]. Approximately 800 women die each day due to childbirth-related complications. Women in developing countries are 36 times more likely to experience pregnancy-related complications compared to those in developed countries [4].

Pregnancies can be terminated safely using WHO-recommended methods, such as medical abortion with medications like mifepristone and misoprostol, or through manual vacuum aspiration (MVA) or dilation and curettage (D&C) performed by trained healthcare providers in appropriate medical settings [5]. Alternatively, pregnancies can be terminated unsafely through self-induced abortions, procedures in unhygienic conditions, or by non-medical practitioners. Globally, Six out of ten unintended pregnancies result in induced abortions, with 97% of unsafe abortions representing about 45% of all abortions occurring in developing nations. Unsafe abortions account for 4.7% to 13.2% of maternal deaths annually. It is estimated that 30 women in developed countries undergo unsafe abortions per 100,000 live births [6].

Preterm birth, a common complication, is the leading cause of death in children under five. Each year, an estimated 15 million babies are born preterm (before 37 weeks of gestation), with preterm birth rates ranging from 5% to 18% of all births, depending on the country. In low-income countries, more than 90% of extremely preterm newborns (born before 28 weeks) die within the first few days of life [7].

Stillbirth occurs when a baby dies after 28 weeks of pregnancy but before or during birth. Nearly 2 million stillbirths occur each year, equating to one every 16 seconds. According to the report "A Neglected Tragedy: The Global Burden of Stillbirths," 84% of these cases occur in low- and lower-middle-income countries. In 2019, three out of four stillbirths occurred in Sub-Saharan Africa or Southern Asia [8].

According to Rwanda's recent demographic and health survey [9], the perinatal mortality rate was 28 deaths per 1,000 pregnancies. This rate includes early neonatal deaths (live births that do not survive past the first 7 days after birth) and stillbirths (pregnancy losses occurring after 7 months of gestation).

Machine learning (ML) has been extensively used to identify potential maternal risks during pregnancy and to forecast childbirth outcomes [10]. In this study, ML algorithms were employed to predict pregnancy outcomes in Rwanda using data from the recent Rwandan demographic and health survey [9]. ML has diverse applications across various fields, including classification and prediction (e.g., pandemic forecasting and medical diagnosis), clustering analysis (e.g., web security through unusual traffic detection, cancer cell identification, and customer segmentation), and natural language processing (e.g., speech recognition, language translation, and sentiment analysis).

Unlike traditional methods that often rely on linear assumptions and limited variable interactions, ML algorithms can handle complex, non-linear relationships and large datasets with numerous predictors, this allows for more accurate and nuanced predictions of pregnancy outcomes [11]. While some researchers have explored forecasting potential pregnancy complications and identifying the optimal delivery method based on maternal features [12], ML has yet to be applied to predict pregnancy outcomes among reproductive women in Rwanda.

## Materials and methods

### Study design

This cross-sectional study utilized data from the 2019–2020 Rwanda Demographic and Health Survey, which provides a comprehensive dataset on various health and demographic factors. The survey collected information from a representative sample of the Rwandan population, enabling the analysis of maternal and pregnancy outcomes within the context of the country's health landscape during this period.

### Study setting

This study was conducted in Rwanda, a small, landlocked country in East Africa with a population of 13,246,394 [13]. Rwanda shares borders with Uganda to the North, Tanzania to the east, Burundi to the south, and the Democratic Republic of Congo to the west. Covering an area of 23,338 square kilometers, Rwanda is one of the most densely populated countries in Africa, with the majority of population concentrated in rural areas (72.1%). Rwanda is known for its remarkable progress in healthcare and socio-economic development since the 1994 genocide, with significant strides made in reducing maternal and child mortality.

### Source of data

The Rwanda Demographic and Health Survey (RDHS) conducted in 2019–2020 was the sixth iteration of its kind, following previous surveys in 1992, 2000, 2005, 2010, and 2014–2015. The National Institute of Statistics of Rwanda (NISR), in collaboration with the Ministry of Health (MOH), was responsible for carrying out this survey. Data collection took place over an extensive period from November 9, 2019, to July 20, 2020. This comprehensive dataset enables the analysis of trends and changes in health and demographic indicators over time, providing valuable insights into maternal and child health in Rwanda. The dataset analyzed was obtained from www.dhsprogram.com after obtaining necessary approval.

### Sampling method

The survey employed a multistage sampling design to ensure accurate estimation of national indicators. This method involved selecting households at each sampling stage to achieve a representative sample size of approximately 13,000 households [9]. For this study, the dataset included information from 14,634 women, which was utilized to investigate predictors of

adverse pregnancy outcomes among women of reproductive age. This robust sample size allowed for a comprehensive analysis of factors associated with pregnancy outcomes and provided valuable insights into maternal health within the Rwandan context.

## Description of variables

**Dependent variable.** Pregnancy outcomes can be classified into several categories, such as live birth (either full-term or preterm), stillbirth, miscarriage, and induced or spontaneous abortion. In this study, the dependent variable was the pregnancy outcome, was derived as a composite variable from multiple existing variables in RDHS. It was created by combining information from variables indicating live births, stillbirths, miscarriages, and abortions. The outcome was then categorized into two primary categories: live birth (including both pre-term and preterm births) and pregnancy loss (Including stillbirths, miscarriages, and abortions. This classification was used to analyze the factors influencing whether a pregnancy resulted in a live birth or a loss. The analysis focused on pregnancies reported in the five years preceding the survey. While we recognize that the loss of an unintended pregnancy may not always be perceived as adverse, for the purposes of this study, any pregnancy loss was considered as such due to its potential health and emotional consequences. While maternal mortality is undoubtedly an adverse outcome, it was not included as a dependent variable in this study due to the focus on pregnancy outcomes specifically related to the fetus or infant.

**Independent variables.** The independent variables, also known as predictors, encompassed a range of sociodemographic information and maternal characteristics of the respondents. These variables included factors such as age, education level, marital status, socioeconomic status, and health-related attributes, which are critical for understanding their influence on pregnancy outcomes.

## Statistical analysis

**Data preprocessing.** The preprocessing involved cleaning the data using various missing value imputation methods, including mean, mode, forward fill, and backward fill. The data were also weighted to ensure that the sample accurately represents the population and to adjust for the complex survey design, including stratification, clustering, and unequal probabilities of selection. To evaluate the model's performance, k-fold cross-validation (with $k = 10$) was employed. This technique ensures a robust and reliable estimate of how well the model will perform on unseen data by splitting the data into ten subsets, training on nine of them, and testing on the remaining one, iteratively. Additionally, the Synthetic Minority Over-sampling Technique (SMOTE) was applied to address class imbalance by generating synthetic examples for the underrepresented class, thereby improving the model's ability to generalize across different classes. All analysis was performed using Python 3.8 and R 4.3.3.

**Univariate analysis.** To describe pregnancy outcomes among reproductive women in Rwanda, a descriptive analysis was conducted among women aged 15–49 years of age. Additionally, a figure was created to visually represent and illustrate the observed pregnancy outcomes based on data from the 2019–2020 Rwanda Demographic and Health Survey.

**Bivariate and multivariate analysis.** To identify the risk factors associated with adverse pregnancy outcomes in Rwanda, bivariate analysis was performed using the chi-square test, in cases where expected cell counts were below 5, Fisher's Exact Test was applied to ensure more accurate results. This analysis assessed the association between independent variables and the dependent variable, with a confidence level of 95% and a significance level of 5%. Associations with p-values less than 0.05 were considered statistically significant. Independent variables that showed significant associations were then included in a logistic regression model to

account for potential confounding variables and to further examine their impact on adverse pregnancy outcomes.

**Machine learning algorithms.**   Machine learning algorithms were employed to predict adverse pregnancy outcomes among women of reproductive age in Rwanda. To address this binary classification problem, a selection of appropriate ML algorithms was utilized. Specifically, seven algorithms were implemented: Logistic Regression, K-Nearest Neighbours (KNN), Decision Trees, Neural Networks, Support Vector Machines (SVM), Random Forest, and Naive Bayes. Each algorithm was evaluated for its effectiveness in predicting the outcomes of interest. Model was trained with 80% and tested with the remaining 20% of the data.

**Model evaluation.**   Each algorithm was evaluated against its metrics which included:

$$\textbf{Accuracy} := \frac{TP + TN}{TP + FP + TN + FN};\tag{1}$$

$$\textbf{Precision} : \frac{TP}{(TP + FP)};\tag{2}$$

$$\textbf{F1 Score} : 2 * \frac{precision * recall}{precision + recall};\tag{3}$$

$$\textbf{Recall score} : \frac{TP}{TP + FN};\tag{4}$$

and **Area under the curve** (AUC) of the receiver operating characteristics (ROC).

Where, TP = true positive; TN = true negative; FP = false positive; and FN = false negative. The ROC curve is a plot of the true positive rate (TPR) against the false positive rate (FPR) at various threshold settings. The Area Under the Curve (AUC) values ranged from 0 to 1, with higher values reflecting better model performance. Additionally, a confusion matrix was generated to assess the deviation between the predicted and actual values, providing insights into the model's accuracy and error rates. The best-performing model was selected based on its evaluation metrics, such as accuracy, precision, recall, and AUC, in comparison to the other models.

**Ethics statement.**   The National Institute of Statistics of Rwanda (NISR) conducted the DHS survey with approval from Rwanda's National Health Research Ethics Committee. Prior to participation, individuals were informed about the study's objectives, procedures, risks, and benefits through various media channels. The consent was obtained from participants before administering the household questionnaires. The analysis was carried out using secondary data provided by the DHS Program.

## Results

### Social demographic characteristics of the respondents

The sociodemographic and maternal characteristics of the respondents were analyzed using descriptive statistics, frequency tables and percentage distribution for each variable are presented in (Table 1).

The results from the 2019–2020 Demographic and Health Survey revealed that among the 14,634 reproductive women surveyed, approximately 22.6% were aged between 15 and 19 years, while only 8.4% were aged 45 to 49 years. Regarding education, 9.2% of respondents had never attended school, more than half (58.1%) had completed primary education, and only 4.6% had received tertiary education. Most women had never been in union (41.4%), 32.2%

**Table 1. Demographic and maternal characteristics of reproductive women in Rwanda.**

| Variables | Frequency (n = 14,634) | Percent (%) |
|---|---|---|
| **Age of respondent** | | |
| 15–19 | 3,308 | 22.6 |
| 20–24 | 2,424 | 16.6 |
| 25–29 | 2,095 | 14.3 |
| 30–34 | 2,047 | 14.0 |
| 35–39 | 2,043 | 14.0 |
| 40–45 | 1,487 | 10.2 |
| 45–49 | 1,230 | 8.4 |
| **Education Level** | | |
| No formal education | 1,352 | 9.2 |
| Primary | 8,500 | 58.1 |
| Secondary | 4,110 | 28.1 |
| Tertiary | 672 | 4.6 |
| **Marital Status** | | |
| Never in union | 6,060 | 41.4 |
| Married | 4,706 | 32.2 |
| Living with partner | 2,584 | 17.7 |
| Separated | 527 | 3.6 |
| Divorced | 379 | 2.6 |
| Widowed | 378 | 2.6 |
| **Religion** | | |
| Catholic | 5,506 | 37.6 |
| Protestant | 6,754 | 46.2 |
| Adventist | 1,842 | 12.6 |
| Muslim | 287 | 2.0 |
| Jehovah's witness | 114 | 0.8 |
| No religion | 106 | 0.7 |
| Other | 24 | 0.2 |
| Traditional | 1 | 0.0 |
| **Residence area** | | |
| Urban | 3,551 | 24.3 |
| Rural | 11,083 | 75.7 |
| **Household head sex** | | |
| Male | 10,045 | 68.6 |
| Female | 4,589 | 31.4 |
| **Occupation** | | |
| Services | 8,176 | 55.9 |
| Not working | 3,955 | 27.0 |
| Unskilled manual | 1,478 | 10.1 |
| Skilled manual | 502 | 3.4 |
| Professional | 409 | 2.8 |
| Clerical | 114 | 0.8 |
| **Wealth status** | | |
| Poorest | 2,844 | 19.4 |
| Poorer | 2,707 | 18.5 |
| Middle | 2,709 | 18.5 |
| Richer | 2,884 | 19.7 |

(*Continued*)

**Table 1.** (Continued)

| Variables | Frequency (n = 14,634) | Percent (%) |
|---|---|---|
| Richest | 3,490 | 23.8 |
| **Distance to Health Facility a problem** | | |
| No | 11,561 | 79.0 |
| Yes | 3,073 | 21.0 |
| **Pregnancy desire** | | |
| Later | 14,046 | 96.0 |
| Not at all | 104 | 0.7 |
| Then | 484 | 3.3 |
| **Age at First Sex** | | |
| Mean ± SD | 14.5 ± 9.5 | |
| **Visited by Health facility** | | |
| Yes | 8,935 | 61.1 |
| No | 5,699 | 38.9 |
| **Number of ANC** | | |
| 0 | 307 | 2.1 |
| 1 | 597 | 4.1 |
| 2 | 1,672 | 11.4 |
| 3 | 5,074 | 34.7 |
| 4 | 6,641 | 45.4 |
| More than 4 | 343 | 2.3 |
| **Smoking** | | |
| Yes | 94 | 0.6 |
| No | 14,540 | 99.4 |
| **Number of unions** | | |
| Mean ± SD | 1.3 ± 0.35 | |
| **Experienced sexual violence** | | |
| Yes | 2,209 | 15.1 |
| No | 12,425 | 84.9 |

Source: Researcher's Analysis of RDHS 2019–2020.

were married and 17.7% were in informal unions. The participants belonged to various religious groups, with approximately 46.2% identifying as Protestants, 37.6% as Catholics, and only a small number practicing traditional beliefs (1).

In terms of residential areas, 75.7% of women lived in rural regions. Employment data showed that more than half (55.9%) were engaged in the services sector, 27% were unemployed at the time of the survey, and 10.1% worked in unskilled manual labor. Wealth status varied, with 23.8% from the wealthiest families, 19.7% from wealthier families, and 18.5% from poorer families. Regarding access to healthcare, 79% of women reported that reaching a health facility was not a significant problem. In terms of pregnancy planning, 96% of women desired to have children later, while 3.3% wanted to become pregnant immediately.

On average, women had their first sexual intercourse at age 14. Healthcare provider visits were reported by 61.1% of women before the survey period. During their pregnancies, 45.4% attended all four recommended antenatal care (ANC) visits, 2.3% attended more than four visits, and 2.1% did not attend any ANC visits. Smoking was reported by only 0.6% of respondents. Women had been in an average of 1.3 unions, and 15.1% of participants had experienced sexual violence

### The pregnancy outcomes among reproductive women in Rwanda

The results indicated that most women who participated in the recent demographic and health surveys delivered live births (93.4%). However, 4.5% of pregnancies ended in miscarriage, and 2.1% resulted in stillbirths (Fig 1).

### Risk factors associated with adverse pregnancy outcome among reproductive women in Rwanda

**Bivariate analysis.** The association between independent variables and diverse pregnancy outcomes (the dependent variable) was evaluated. Both chi-square statistics and p-values were computed, with a p-value less than 0.05 indicating a statistically significant association (Table 2).

The findings revealed that most women, regardless of age, delivered live births. However, a higher proportion of adverse pregnancy outcomes was observed among women aged 35 years and older. Specifically, 20.4% of women aged 35–39, 25.3% of those aged 40–44, and 27.2% of women aged 45–49 experienced adverse pregnancy outcomes. Age was significantly associated with adverse pregnancy outcomes (p < 0.001).

Educational level was not significantly associated with adverse pregnancy outcomes (p = 0.106), although a higher proportion of adverse outcomes among women with no formal education (18.8%). Marital status was significantly linked to adverse pregnancy outcomes (p < 0.001), with married women experiencing more adverse outcomes (22.8%). Religion was not associated with adverse pregnancy outcomes (p = 0.600), with a higher proportion observed among women with no religious affiliation (18.9%), followed by those in Protestant denominations (12.9%).

Residential location was not associated with adverse pregnancy outcomes (p = 0.894), with 12.4% of women in rural areas experiencing adverse outcomes compared to 11.1% in urban areas. The gender of the household head also not correlated with adverse pregnancy outcomes (p = 0.331), with male-headed households showing a higher rate of adverse outcomes (13.5%). The occupation of respondents was not associated with adverse pregnancy outcomes (p = 0.062), with a higher proportion observed among women working in the services sector (16.7%) compared to those in skilled manual jobs (3.8%).

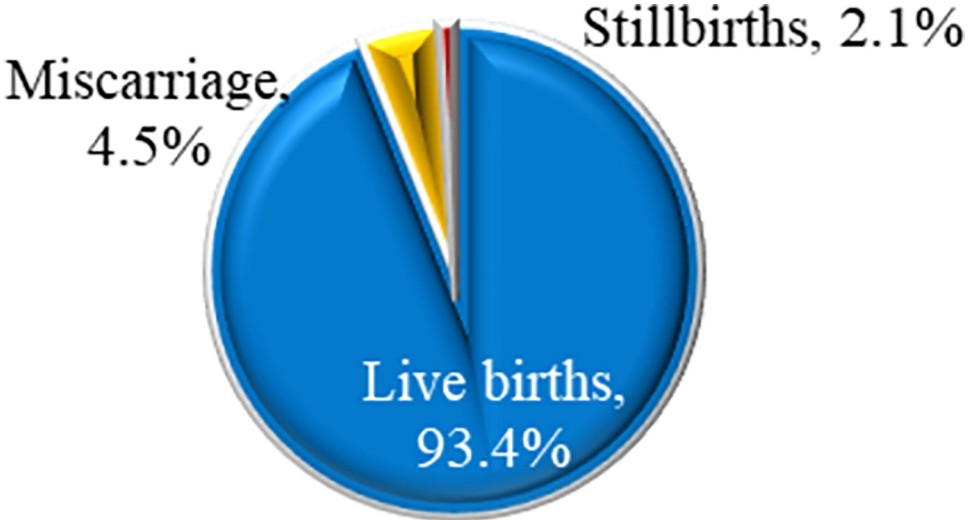

**Fig 1. Pregnancy outcomes among reproductive women in Rwanda.**

**Table 2. Bivariate analysis: Factors associated with adverse pregnancy outcome among reproductive women in Rwanda.**

| Variables | Adverse pregnancy outcome | | Chi-square/Fisher's Exact | p-value |
|---|---|---|---|---|
| | No (%) | Yes (%) | | |
| **Age of respondent** | | | | |
| 15–19 | 99.6 | 0.4 | 1216.220 | **<0.001** |
| 20–24 | 95.6 | 4.4 | | |
| 25–29 | 89.4 | 10.6 | | |
| 30–34 | 85.8 | 14.2 | | |
| 35–39 | 79.6 | 20.4 | | |
| 40–45 | 74.7 | 25.3 | | |
| 45–49 | 72.8 | 27.2 | | |
| **Education Level** | | | | |
| No formal education | 81.2 | 18.8 | 178.147 | **0.106** |
| Primary | 86.5 | 13.5 | | |
| Secondary | 93.1 | 6.9 | | |
| Tertiary | 88.2 | 11.8 | | |
| **Marital Status** | | | | |
| Never in union | 98.8 | 1.2 | | |
| Married | 77.2 | 22.8 | 1,244.597 | **<0.001** |
| Living with partner | 83.7 | 16.3 | | |
| Separated | 85 | 15.0 | | |
| Divorced | 87.3 | 12.7 | | |
| Widowed | 82.3 | 17.7 | | |
| **Religion** | | | | |
| Catholic | 89.2 | 10.8 | 20.050(Fisher's Exact) | **0.600** |
| Protestant | 87.1 | 12.9 | | |
| Adventist | 87.3 | 12.7 | | |
| Muslim | 89.2 | 10.8 | | |
| Jehovah's witness | 89.5 | 10.5 | | |
| No religion | 81.1 | 18.9 | | |
| Other | 95.8 | 4.2 | | |
| Traditional | 100 | 0.0 | | |
| **Residence area** | | | 4.128 | **0.089** |
| Urban | 88.9 | 11.1 | | |
| Rural | 87.6 | 12.4 | | |
| **Household head sex** | | | | |
| Male | 86.5 | 13.5 | 60.7 | **0.331** |
| Female | 91.1 | 8.9 | | |
| **Occupation** | | | 275.707 | **0.062** |
| Services | 83.3 | 16.7 | | |
| Not working | 93.9 | 6.1 | | |
| Unskilled manual | 86.6 | 13.4 | | |
| Skilled manual | 96.2 | 3.8 | | |
| Professional | 84.4 | 15.6 | | |
| Clerical | 89.5 | 10.5 | | |
| **Wealth status** | | | | |

*(Continued)*

**Table 2.** (Continued)

| Variables | Adverse pregnancy outcome | | | |
|---|---|---|---|---|
| | No (%) | Yes (%) | Chi-square/Fisher's Exact | *p*-value |
| Poorest | 87.9 | 12.1 | 5.595 | 0.231 |
| Poorer | 88.4 | 11.6 | | |
| Middle | 86.8 | 13.2 | | |
| Richer | 87.7 | 12.3 | | |
| Richest | 88.7 | 11.3 | | |
| **Distance to Health Facility** | | | | |
| No problem | 88.3 | 11.7 | 7.624 | **0.205** |
| Problem | 86.5 | 13.5 | | |
| **Pregnancy desire** | | | | |
| Later | 88.3 | 11.7 | 51.065 | **0.090** |
| Not at all | 79.8 | 20.2 | | |
| Then | 78.3 | 21.7 | | |
| **Age at First Sex** | | | | |
| Mean | 13.63 | 20.46 | 28.99 | **<0.001** |
| **Visited by Healthcare provider** | | | | |
| Yes | 85.1 | 14.9 | 180.643 | **<0.001** |
| No | 92.5 | 7.5 | | |
| **Number of ANC** | | | | |
| 0 | 89.3 | 10.7 | 18.704 | **0.002** |
| 1 | 91.0 | 9.0 | | |
| 2 | 88.8 | 11.2 | | |
| 3 | 87.4 | 12.6 | | |
| 4 | 87.8 | 12.2 | | |
| More than 4 | 81.9 | 18.1 | | |
| **Smoking** | | | | |
| Yes | 88.3 | 11.7 | 0.000 | 0.989 |
| No | 87.9 | 12.1 | | |
| **Number of unions** | | | | |
| Mean | 1.28 | 1.13 | 17.367 | **<0.001** |
| **Experienced sexual violence** | | | | |
| Yes | 87.4 | 12.6 | 0.393 | 0.530 |
| No | 88.0 | 12.0 | | |

Source: Researcher's Analysis of RDHS 2019–2020.

Distance to health facilities was not linked to adverse pregnancy outcomes (p = 0.205), despite a higher proportion of adverse pregnancies among women who reported difficulty accessing health facilities (13.5%). Pregnancy desire was significantly associated with adverse outcomes (p < 0.001), with more adverse pregnancies observed among women who desired pregnancy (21.7%). Age at first sexual intercourse was significantly associated with adverse pregnancy outcomes (p < 0.001), with women experiencing adverse outcomes starting their sexual activity later (average age 20.5 years) compared to those who started earlier (average age 13.6 years).

Being visited by a health care provider was significantly associated with adverse pregnancy outcomes (p < 0.001). ANC utilization was also significant (p = 0.002), with a higher proportion of adverse outcomes observed among women who attended more than four ANC visits

(18.1%). Finally, the number of unions a woman had been in was significantly associated with adverse pregnancy outcomes (p < 0.001).

## Multivariable analysis of factors associated with adverse pregnancy outcome among reproductive women in Rwanda

To explore risk factors while adjusting for confounders and effect modification, binary logistic regression was performed. The analysis calculated the adjusted odds ratios (AOR), 95% confidence intervals, and p-values for variables that demonstrated significant relationships in the bivariate analysis. This approach allowed for a more nuanced understanding of the associations between predictors and adverse pregnancy outcomes, accounting for potential confounding variables and interactions (Table 3).

The findings revealed that the respondents' age was significantly associated with adverse pregnancy outcomes. As age increased, so did the risk of adverse pregnancy outcomes. For

**Table 3. Multivariate analysis of risk factors associated with adverse pregnancy outcome among reproductive women in Rwanda.**

| Variables | Adverse pregnancy outcome | | | |
|---|---|---|---|---|
| | AOR | [95% conf. interval] | | *p*-value |
| **Age of respondent** | | | | |
| 15–19 | 1* | | | |
| 20–24 | 3.452 | 0.909 | 6.680 | 0.062 |
| 25–29 | 4.953 | 1.167 | 9.371 | 0.104 |
| 30–34 | 5.755 | 3.085 | 10.074 | <**0.001** |
| 35–39 | 8.458 | 4.507 | 10.571 | <**0.001** |
| 40–44 | 11.86 | 6.250 | 21.842 | <**0.001** |
| 45–49 | 14.233 | 7.359 | 25.922 | <**0.001** |
| **Marital status** | | | | |
| Never in union | 1* | | | |
| Married | 0.894 | 0.787 | 0.966 | **0.011** |
| Living with partner | 1.651 | 4.473 | 8.130 | 0.061 |
| Separated | 4.428 | 3.199 | 6.684 | 0.104 |
| Divorced | 3.082 | 2.126 | 4.895 | 0.246 |
| Widowed | 3.524 | 2.453 | 5.443 | 0.820 |
| **Age at first sex** | | | | |
| | 1.027 | 0.628 | 1.341 | **0.061** |
| **Visited by a healthcare provider** | | | | |
| Yes | 1* | | | |
| No | 1.421 | 1.3 | 1.611 | <**0.001** |
| **Number of ANC visits** | | | | |
| 0 | 1* | 0.510 | 1.185 | 0.275 |
| 1 | 0.973 | 0.732 | 1.277 | 0.848 |
| 2 | 0.801 | 0.664 | 0.961 | **0.019** |
| 3 | 0.945 | 0.838 | 1.065 | 0.357 |
| 4 | 0.791 | 0.510 | 1.185 | 0.275 |
| More than 4 | 1.534 | 1.096 | 2.117 | **0.071** |
| **Number of unions** | | | | |
| | 1.320 | 1.104 | 1.573 | **0.002** |

Source: Researcher's Analysis of RDHS 2019–2020

women aged 30–34 years, the risk was more than 5 times higher (AOR = 5.755, 95% CI = 3.085–10.074, p < 0.001), and for those aged 35–39 years, it was more than 8 times higher (AOR = 8.458, 95% CI = 4.507–10.571, p < 0.001). The risk was around 12 times higher for women aged 40–44 years (AOR = 11.86, 95% CI = 6.250–21.842, p < 0.001), and more than 14 times higher for those aged 45–49 years (AOR = 14.233, 95% CI = 7.359–25.922, p < 0.001), compared to women aged 15–19 years.

Marital status also significantly influenced adverse pregnancy outcomes. Married women were 0.9 times less likely to experience adverse outcomes compared to their unmarried counterparts (AOR = 0.894, 95% CI = 0.787–0.966, p = 0.011). The women who were not visited by a healthcare provider during pregnancy were more likely to experience advserse pregnancy outcome (AOR = 1.421, 95% CI = 1.300–1.611).

Antenatal care (ANC) utilization showed a protective effect. Women who attended at least two ANC visits were 0.8 times less likely to experience adverse pregnancy outcomes compared to those who did not attend any ANC visits (AOR = 0.801, 95% CI = 0.664–0.9615, p = 0.019). Finally, the number of unions a woman had experienced was positively associated with adverse pregnancy outcomes. Each additional union increased the likelihood of adverse outcomes by 1.32 times (AOR = 1.320; 95% CI = 1.104–1.573; p = 0.002).

## Model performance evaluation

The K-Nearest Neighbors (KNN) model demonstrated a strong performance with a recall score of 0.97, indicating its high ability to correctly identify most instances of adverse pregnancy outcomes. Its precision score of 0.89 also reflects a good capability in correctly identifying positive cases. The AUC score of 0.842 for KNN suggests that it is highly capable of distinguishing between different outcome categories, making it one of the top-performing models.

The Decision Tree model exhibited a recall score of 0.95 and a precision score of 0.88. This shows that it is effective at identifying adverse pregnancy outcomes and accurately classifying positive instances. Despite these strong scores, the Decision Tree's F1 and AUC scores were slightly lower than those of the KNN model and the Random Forest, indicating that its overall performance might be less robust. The Random Forest model achieved an AUC score of 0.81, which, while higher than that of the Naïve Bayes model, was lower than the AUC scores for Logistic Regression, Support Vector Machine (SVM), Neural Network, and KNN models. Despite this, the Random Forest model's accuracy, precision, recall, and F1 scores were commendable, underscoring its effectiveness in predicting adverse pregnancy outcomes, though it did not reach the highest performance levels of the top models.

The KNN model highlighted that woman with advanced maternal age (>30 years), those in multiple unions, and those who were not visited by a health care provider were more prone to adverse pregnancy outcomes. Additionally, women who lacked adequate antenatal care (ANC) visits and unmarried women were identified as being at higher risk. These findings are consistent with the results from the logistic regression model, further validating the predictive power of machine learning in identifying high-risk categories for adverse pregnancy outcomes.

Overall, the KNN, Decision Tree, and Random Forest models demonstrated high AUC scores relative to other models, marking them as effective tools for predicting adverse pregnancy outcomes (Table 4 and **Fig 2**).

## Discussion

The results from our study revealed that, during the five years preceding the 2019–2020 Rwanda Demographic and Health Survey, 93.4% of pregnancies resulted in live births, 4.5%

**Table 4. Overall model performance.**

| ML model | Accuracy | Precision | Recall score | F1 score | AUC |
|---|---|---|---|---|---|
| Logistic regression | 0.879 | 0.88 | 1.00 | 0.94 | 0.77 |
| Support vector machine | 0.88 | 0.88 | 1.00 | 0.94 | 0.77 |
| KNN model | 0.86 | 0.89 | 0.97 | 0.93 | 0.842 |
| Decision tree | 0.85 | 0.88 | 0.95 | 0.92 | 0.846 |
| Neural network | 0.88 | 0.88 | 1.00 | 0.94 | 0.77 |
| Naïve Bayes | 0.75 | 0.92 | 0.78 | 0.85 | 0.76 |
| Random Forest | 0.81 | 0.88 | 0.90 | 0.82 | 0.81 |

ended in miscarriages, and 2.1% ended in stillbirths. These findings align with the recent Rwanda Demographic and Health Survey, which reported that 21 out of every 1,000 pregnancies end in stillbirths. Notably, the miscarriage rate in our study was 20% lower than the global estimate, indicating that Rwanda is making progress in reducing pregnancy loss rates through improvements in maternal health services [9].

Our analysis also highlighted that age was a significant factor associated with adverse pregnancy outcomes. As the age of respondents increased, so did the risk of experiencing adverse pregnancy outcomes. This finding is consistent with a study in China which has shown that pregnancy at the age of 35 years or older, known as advanced maternal age (>30 years), is a notable risk factor for negative pregnancy and childbirth outcomes compared to younger women [14].

Our study found that marital status significantly influenced adverse pregnancy outcomes. Compared to single women, married women were less likely to experience adverse pregnancy

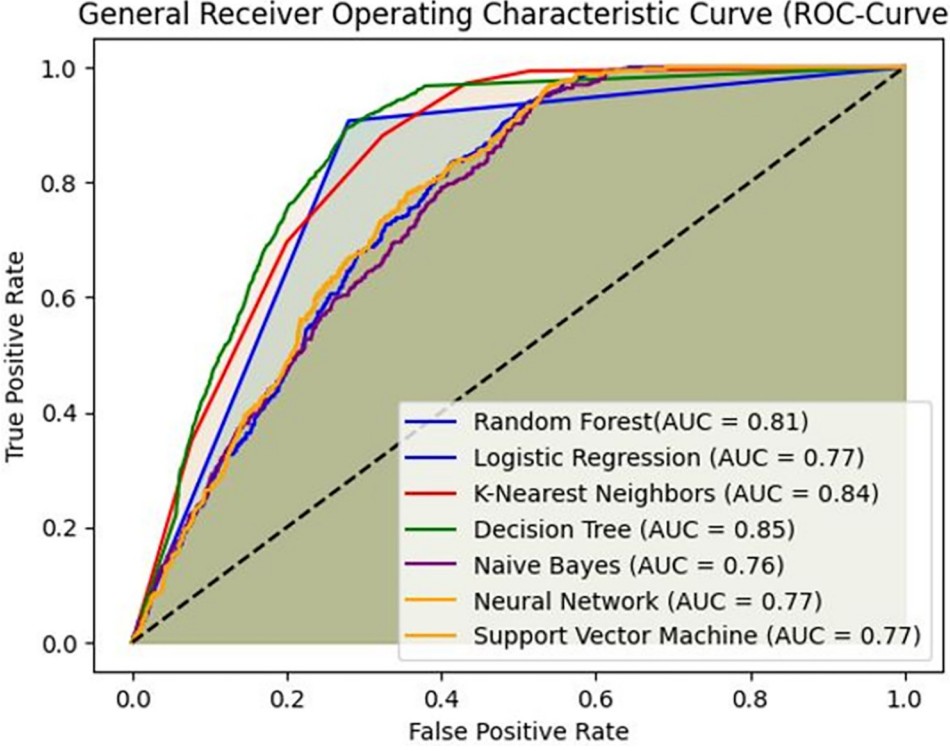

**Fig 2. Overall model performance.**

outcomes. This finding contrast with research conducted in the United States, which reported that unmarried mothers had a higher risk of infant mortality compared to married mothers, even when accounting for factors such as maternal age and education [15], the difference could be explained by differences in population demographics, healthcare access and cultural norms. Married women may benefit from increased social support and engagement in healthier behaviours, such as regular prenatal care, a nutritious diet, and exercise, which can mitigate the risk of adverse outcomes. In contrast, unmarried women may face additional stressors, including social stigma and lack of support, which can negatively impact pregnancy outcomes [16].

Our results also demonstrated that women who were visited by healthcare provider during pregnancy were less likely to encounter adverse pregnancy outcomes. Moreover, women who attended two or more antenatal care (ANC) visits were 0.8 times less likely to experience adverse pregnancy outcomes compared to those who did not attend any ANC visits. This finding aligns with a study conducted in Ghana, which indicated that regular attendance at ANC appointments and receiving appropriate prenatal care significantly improved pregnancy outcomes, including reduced risks of maternal and infant mortality, preterm birth, and low birth weight [17].

Additionally, our study revealed that the number of unions a woman has had was significantly associated with adverse pregnancy outcomes. Women with more unions were more likely to experience adverse pregnancy outcomes. This is supported by studies suggesting that women with multiple unions may face higher risks of preterm birth, low birth weight, and infant mortality compared to those with fewer unions [18]. Women with multiple unions might engage in behaviors that increase these risks, such as smoking, substance abuse, or risky sexual behavior, which can negatively affect both maternal and fetal health.

The overarching goal of this study was to predict adverse pregnancy outcomes using supervised machine learning algorithms. We employed and evaluated seven machine learning algorithms: decision trees, neural networks, random forests, K-nearest neighbors (KNN), naïve Bayes, logistic regression, and support vector machines (SVM). Among these, the KNN, decision tree, and random forest models demonstrated high AUC scores and were effective in predicting adverse pregnancy outcomes.

This study has significant implications for research, policy, and practice. In research, the findings demonstrate the potential of machine learning algorithms to predict adverse pregnancy outcomes, suggesting the need for future studies to validate these models using diverse datasets and real-time clinical data. In terms of policy, the study highlights the importance of strengthening maternal health programs, especially by expanding access to antenatal care (ANC) services and targeting high-risk groups such as older women and those with multiple unions. From a practice standpoint, healthcare providers can use machine learning tools to improve risk stratification and provide tailored care to pregnant women, ensuring early interventions and better maternal and neonatal outcomes.

## Conclusion

The findings revealed that 4.5% of pregnancies ended in miscarriage and 2.1% in stillbirths. Key risk factors for adverse pregnancy outcomes included advanced maternal age (>30 years), being unmarried, no visit by a healthcare provider, inadequate antenatal care utilization, and having multiple unions. Among the machine learning models evaluated, the K-Nearest Neighbors (KNN) algorithm demonstrated superior performance, achieving the highest predictive accuracy for adverse pregnancy outcomes. While these results suggest that KNN is a promising tool for early risk identification, further research is needed to validate these findings across

broader populations and settings before machine learning models can be integrated into clinical practice for maternal healthcare in Rwanda.

## Limitation of the study

This study has some limitations. First, the cross-sectional design limits the ability to establish causality between identified risk factors and adverse pregnancy outcomes. Additionally, the use of self-reported data from the RDHS may lead to recall bias or inaccuracies. Our dependent variable, which categorized outcomes as either live birth or pregnancy loss, does not account for the complexity of whether the pregnancy was desired, pregnancy loss may not always be considered adverse in the case of an unwanted pregnancy. Furthermore, maternal mortality and live birth complications were not included in the analysis, which could provide a more comprehensive view of pregnancy outcomes. Lastly, while machine learning models demonstrated predictive capabilities, their performance could be improved with more detailed clinical data and external validation using independent datasets.

## Study contribution

The study provides valuable insights into the use of machine learning techniques to predict adverse pregnancy outcomes. Offering an innovative approach compared to traditional statistical methods. By applying models likes K-nearest neighbors (KNN) and validating their performance, the study demonstrates how data-driven techniques can improve the accuracy of predictions related to maternal health, particularly in resource-limited settings. The study contributed to the understanding of the specific risk factors, such as advanced maternal age, marital status, and ANC attendance, in relation to adverse pregnancy outcomes in Rwanda. This context-specific analysis provides policy makers wand healthcare provides with actionable information that could help target interventions more effectively to reduce maternal and neonatal morbidity and mortality.

## Acknowledgments

We would like to thank the African Center of Excellence in Data Science at the University of Rwanda for their enthusiasm and support. We are also deeply grateful to our families and friends for their encouragement throughout this work.

## Author Contributions

**Conceptualization:** Theogene Kubahoniyesu.

**Formal analysis:** Theogene Kubahoniyesu.

**Investigation:** Theogene Kubahoniyesu.

**Methodology:** Ignace Habimana Kabano.

**Supervision:** Ignace Habimana Kabano.

**Writing – original draft:** Theogene Kubahoniyesu.

**Writing – review & editing:** Theogene Kubahoniyesu.

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
