## [Decision Letter · Decision Letter 0]

20 Aug 2024

PONE-D-23-34183Predicting adverse pregnancy outcome in Rwanda using machine learning techniquesPLOS ONE

Dear Dr. KUBAHONIYESU,

Thank you for submitting your manuscript to PLOS ONE. After careful consideration, we feel that it has merit but does not fully meet PLOS ONE’s publication criteria as it currently stands. Therefore, we invite you to submit a revised version of the manuscript that addresses the points raised during the review process.

**ACADEMIC EDITOR: **

The manuscript has been reviewed and would require further improvement. A revision should be made based on the reviewers' comments. ==============================

We look forward to receiving your revised manuscript.

Kind regards,

Obasanjo Afolabi Bolarinwa

Academic Editor

PLOS ONE

Reviewers' comments:

Reviewer's Responses to Questions

**Comments to the Author**

1. Is the manuscript technically sound, and do the data support the conclusions?

Reviewer #1: Partly

Reviewer #2: Partly

2. Has the statistical analysis been performed appropriately and rigorously? 

Reviewer #1: I Don't Know

Reviewer #2: No

3. Have the authors made all data underlying the findings in their manuscript fully available?

Reviewer #1: No

Reviewer #2: Yes

4. Is the manuscript presented in an intelligible fashion and written in standard English?

Reviewer #1: No

Reviewer #2: No

5. Review Comments to the Author

Reviewer #1: This study predicted the adverse pregnancy outcome in Rwanda using Machine Learning Techniques. Below are few observation to enrich the quality of the manuscript.

Abstract:

In the abstract where you stated 14,634 women, 93.4% live birth etc. what do you mean by 4.5....is it for miscarriage or still birth?? or both??. 2.1% is for what??

Live with more than one union?? means what? Do you mean polyandry?

Mothers who attend ANC---- Who are they? single or married?

Key words should be at least 5

Introduction:

Line 1-4 is lengthy. Make your sentence short and precise. Line 3-5 provide the sources of this statement (86% of global maternal fatalities in year 2017 occurs in SSA and Southern Asian. Also 1/5th occurs in Southern Asia. Provide sources of these statement

The first statement is clumsy and need contextual editing for clarity

Paragraph 2 is clumsy and not clear

The entire introductory section is not clear and difficult to comprehend the direction of the authors. Please restructure.

Poor English Language

Method

What is the method use for cleaning data. Please check again and revert

Data was collected from November 9, 2019 to July 20, 2020---This is a secondary data that could be downloaded within a minute. There is no need for the date as data was not collected by you. Please provide approval certificate for the use of this dataset.

Dependent Variable:

1: Pregnancy loss: How do you determine that a pregnancy loss was an adverse outcome? Loss of unwanted pregnancy could be a positive outcome for both the man and the woman. While live birth with complications could be an adverse outcome. Kindly look into these for clarity)

2. Live Birth

What about maternal mortality, is that not adverse effect too?

Kindly check the data processing of missing variable to be sure its accurate

Univariate

What is the age range of reproductive women under consiration. Clarity is needed for mothers (single or married)

Discussion

You stated that data was for 2029-2020. Please check and revert. What is the position of this study, i cant find it in the discussion unit

Result

This section is not clear

This need thorough editing for English and context.

Reviewer #2: You have only shown the risk factors for adverse outcomes as statistical regressions would. You did no ML predictions. What you stopped at could only have been justified if the logistic regression was your highest performing model.

6. PLOS authors have the option to publish the peer review history of their article (what does this mean?). If published, this will include your full peer review and any attached files.

Reviewer #1: No

Reviewer #2: **Yes: **Tosin Olajide Oni

---

## [Author Response · Author response to Decision Letter 0]

5 Sep 2024

Dear Reviewers

Thank you for the opportunity to revise and resubmit our manuscript titled "Predicting Adverse Pregnancy Outcome in Rwanda Using Machine Learning Techniques" (Manuscript ID: PONE-D-23-34183). We appreciate the constructive feedback provided by the reviewers and have made the necessary revisions to address all concerns.

Thanks

---

## [Decision Letter · Decision Letter 1]

27 Sep 2024

PONE-D-23-34183R1Predicting adverse pregnancy outcome in Rwanda using machine learning techniquesPLOS ONE Dear Dr. KUBAHONIYESU,

Thank you for submitting your manuscript to PLOS ONE. After careful consideration, we feel that it has merit but does not fully meet PLOS ONE’s publication criteria as it currently stands. Therefore, we invite you to submit a revised version of the manuscript that addresses the points raised during the review process.

We look forward to receiving your revised manuscript.

Kind regards,

Obasanjo Afolabi Bolarinwa

Academic Editor

PLOS ONE

Journal Requirements:

Reviewers' comments:

Reviewer's Responses to Questions

**Comments to the Author**

1. If the authors have adequately addressed your comments raised in a previous round of review and you feel that this manuscript is now acceptable for publication, you may indicate that here to bypass the “Comments to the Author” section, enter your conflict of interest statement in the “Confidential to Editor” section, and submit your "Accept" recommendation.

Reviewer #1: All comments have been addressed

Reviewer #2: (No Response)

2. Is the manuscript technically sound, and do the data support the conclusions?

Reviewer #1: Partly

Reviewer #2: Yes

3. Has the statistical analysis been performed appropriately and rigorously? 

Reviewer #1: Yes

Reviewer #2: No

4. Have the authors made all data underlying the findings in their manuscript fully available?

Reviewer #1: Yes

Reviewer #2: Yes

5. Is the manuscript presented in an intelligible fashion and written in standard English?

Reviewer #1: Yes

Reviewer #2: Yes

6. Review Comments to the Author

Reviewer #1: This is an improvement to earlier submission. Great job.

Kindly take note of below

1. Key words: Please add Neonatal mortality, Maternal mortality

2. Edit the work for English and context. Look at the introduction---2nd line in 2nd paragraph. Poor English. This is also seen in other places too. Kindly proofread the work

3. Discussion: state the limitation of the study. Also add the policy implication of the study.

4. contribution of the study to knowledge

Reviewer #2: Direct quotes from the manuscript are in red

Thanks to the authors for taking their time to attend to the reviewers’ comments. However, it would have been expected that a point-by-point response to the comments would be uploaded, as is the standard, alongside the reviewed manuscript.

Introduction

Lines 59 - 60: please cite your source

Lines 73 – 74: really? How are pregnancy outcomes predictable by mothers? These are not in the ‘3’ you cited

Line 83: what are WHO-recommended methods?

Line 122: The strengths of ML over the traditional statistical methods that have predominantly been used may be a better justification.

Dependent variable

The re-categorization of the dependent variable makes sense now. However, since the variable “pregnancy outcome’ does not exist in the DHS, it is advised that the authors specify how the variable was obtained i.e. whether it is a composite gotten from a combination of other variables, an operationalized term from another variable in the DHS, or a simulation. Also, authors need to state the time reference, i.e. does this refer to the women’s pregnancy in the last one or five years?, This is important for open science (verifiability and reproducibility), which is the essence of the methodology section.

Independent variables

Authors need to provide descriptions of the independent variables. For instance, the results show that “…woman with 48 advanced maternal age, those in multiple unions, and those who delayed their first sexual intercourse, who lacked adequate ANC were more prone to adverse pregnancy outcomes…”

Readers may wonder what ‘advanced maternal age/ means? Please describe your variables or provide citations if you adopted existing measurements. Please do this for variables that are not straight forward.

Machine learning algorithms

“Model was trained with 20% and tested with the remaining 80% of the data”

Really? I guess this was a mistake. Premising prediction on 20% training must only have given spurious results. Training must take at least 50%, even up to 80% in many cases.

Discussion

The results show that “The KNN model highlighted that women with advanced maternal age, those in multiple unions, 349 and those who delayed their first sexual intercourse were more prone to adverse pregnancy 350 outcomes”

There is overwhelming evidence in the literature that delaying first sexual intercourse (unlike exposure before age 18) is associated with positive maternal outcomes. If the result of this study shows otherwise, authors need to spend time explaining why…and the implications of this on abstinence advocacy.

Please attach a point-by-point response If this manuscript is sent back to me for review.

7. PLOS authors have the option to publish the peer review history of their article (what does this mean?). If published, this will include your full peer review and any attached files.

Reviewer #1: **Yes: **Chukwudeh Stephen Okechukwu

Reviewer #2: No

---

## [Author Response · Author response to Decision Letter 1]

5 Oct 2024

We would like to thank you for your thorough review of our manuscript (ID: PONE-D-23-34183R1), titled “Predicting Adverse Pregnancy Outcome in Rwanda Using Machine Learning Techniques”, submitted to PLOS ONE. We appreciate the constructive comments and suggestions, which have greatly helped us improve the quality of our work. we provided a point-by-point response to each of the reviewers' and editor's comments, indicating the revisions made to the manuscript.

---

## [Editor Report · Decision Letter 2]

8 Oct 2024

Predicting adverse pregnancy outcome in Rwanda using machine learning techniques

PONE-D-23-34183R2

Dear Dr. Kubahoniyesu,

We’re pleased to inform you that your manuscript has been judged scientifically suitable for publication and will be formally accepted for publication once it meets all outstanding technical requirements.

Kind regards,

Obasanjo Afolabi Bolarinwa

Academic Editor

PLOS ONE

Additional Editor Comments (optional):

All comments have been addressed.
---

## [Editor Report · Acceptance letter]

17 Oct 2024

PONE-D-23-34183R2 

PLOS ONE

Dear Dr. KUBAHONIYESU, 

I'm pleased to inform you that your manuscript has been deemed suitable for publication in PLOS ONE. Congratulations! Your manuscript is now being handed over to our production team.

Kind regards, 

on behalf of

Mr Obasanjo Afolabi Bolarinwa 

Academic Editor

PLOS ONE